# Converting Bilateral Free-End Removable Partial Dentures to Implant-Assisted Removable Partial Dentures Using 6 mm Short Implants: Patient-Reported Outcomes of a Prospective Clinical Study

**DOI:** 10.3390/ijerph19158998

**Published:** 2022-07-24

**Authors:** Samir Abou-Ayash, Anne-Carole Rudaz, Simone Janner, Dominik Kraus, Martin Schimmel, Norbert Enkling

**Affiliations:** 1Department of Reconstructive Dentistry and Gerodontology, School of Dental Medicine, University of Bern, 3010 Bern, Switzerland; anne-carole.r@bluewin.ch (A.-C.R.); or s.janner@ziko-bern.ch (S.J.); martin.schimmel@unibe.ch (M.S.); or enkling@uni-bonn.de (N.E.); 2Surgery Center ZIKO, 3008 Bern, Switzerland; 3Department of Oral Surgery, University Center for Dental Medicine Basel, University of Basel, 4058 Basel, Switzerland; 4Department of Prosthodontics, Preclinical Education and Dental Materials Science, University of Bonn, 5311 Bonn, Germany; dominik.kraus@ukbonn.de; 5Division of Gerodontology and Removable Prosthodontics, University of Geneva, 1205 Geneva, Switzerland

**Keywords:** OHIP, oral health-related quality of life, short implants, patient-reported outcomes, patient satisfaction, abutment type, removable partial denture, RPD

## Abstract

The study assessed oral health-related quality of life (OHRQoL) of patients who received two 6 mm short implants in mandibular molar sites, converting existing bilateral free-end removable partial dentures (RPDs) to implant-assisted RPDs (IARPDs). After a postsurgical healing period of 4 months, the participants received a non-retentive dome abutment for 8 weeks, and then a retentive ball abutment for another 8 weeks. Afterwards, the participants made their final choice on which abutment to keep. The final follow-up was 1 year after implant placement. OHRQoL was evaluated with the 49-items version of the Oral Health Impact Profile (OHIP-49) at the abutment exchanges and the final follow-up. Furthermore, numerical rating scales were used to analyze patient satisfaction after 1 year. Questionnaire data of 13 participants were evaluated. Overall, OHRQoL increased with both the dome (*p* = 0.02) and the ball abutments (*p* < 0.001), without a significant difference between the abutments (*p* = 0.953). The questionnaires revealed an improvement in terms of oral situation, quality of life, and masticatory capacity (all *p* < 0.01). Patients showed a significant preference for the ball abutments (*p* < 0.001). Converting RPDs to IARPDs resulted in significant improvement of OHRQoL. Patients seem to prefer retentive over non-retentive abutments, although no differences in terms of OHRQoL were observed.

## 1. Introduction

Although public health programs have enabled people to keep their teeth until an advanced age or even their whole lives, the loss of single or multiple teeth remains a common event [1]. Therefore, replacement of lost teeth remains a reality which dentists are confronted with in daily practice [2].

The most simple and economical treatment for the rehabilitation of occlusion and function in subjects with partial edentulism is a clasp-retained, removable partial denture (RPD) [3]. However, various studies demonstrated lower masticatory performance in RPD patients, compared to patients rehabilitated by fixed partial dentures [4]. This may be caused by rotational movements often found in clasp-retained RPDs, especially in the ones with uni- or bilateral free-end-saddles (Kennedy Class I and II) [5,6]. Together, Kennedy Class I and Class II edentulism account for almost one-third of all partial edentulous situations [7]. The lack of stability of such prostheses is therefore a problem with an important stake.

The loss of post-canine teeth leads to reduced occlusal force and a decline in masticatory performance [8]. Various studies have demonstrated that implant placement in patients treated with free-ended RPDs is a valid option, converting the RPDs to implant-assisted RPD (IARPDs) [9,10]. The beneficial effects of implant placement include improved mastication and nutrient intake [11], increased maximal bite force, comminution food index, and masseter muscle thickness during contraction [12]. However, implant placement in posterior mandibular sites, especially in presence of severe atrophy, can be challenging or even impossible due to the resulting proximity to the mandibular canal or the sublingual fossa. Consequently, a trend towards placing short implants with large diameters has appeared in recent years [13], thus avoiding additional complex and costly bone augmentation procedures [14]. These types of implants have demonstrated similar short and mid-term success and survival rates when compared to longer implants [15].

In addition to improving masticatory performance, implant placement also increases patient satisfaction [16], even more when placed in the molar region [17]. Assessing the quality of care and patient satisfaction is of great significance, especially in the dental field, where patient survival is not at stake [18,19]. For this purpose, there are several validated patient-reported outcome measures (PROMs) among which the Oral Health Impact Profile (OHIP) is one of the most frequently applied measure in dentistry is [20], evaluating the Oral Health-Related Quality of Life (OHRQoL). When converting bilateral free-end RPDs (Kennedy Class I) to Kennedy Class III IARPDs, an increase of OHRQoL has been demonstrated [21,22], but only a few studies evaluated different types of abutments in IARPDs. Therefore, further clinical studies are needed to evaluate the influence of the attachment type on clinical and patient-reported outcomes, especially when short implants are used [15,23,24].

This prospective clinical study analyzed patient-reported outcomes, using two types of implant abutments in patients with Kennedy class I RPDs that had been converted to IARPDs, by the use of 6 mm long implants that were placed bilaterally in mandibular molar sites. The first 0-hypothesis (H-01) was that the placement of a non-retentive implant abutment would not impact the OHRQoL. The second 0-hypothesis (H-02) was that there would be no difference between the two abutments in terms of OHRQoL. Furthermore, the patients’ final abutment choice as well as their satisfaction after completing the treatment were evaluated.

## 2. Materials and Methods

The study was designed as a prospective clinical study, evaluating two types of attachments on 6 mm short implants in the posterior mandible supporting mandibular IARPDs, in terms of patient-reported outcomes. It was conducted according to the standards of the Declaration of Helsinki. Ethical approval was granted by the Cantonal Ethics Committee Bern (CEC; No. 223/13). The study was registered in the German Clinical Trials Register (DRKS; Number: DRKS00024147). All included participants signed the informed consent form. The subjects were recruited at the Department of Prosthodontics, School of dentistry, University Bern, Switzerland. Volunteers, wearing sufficient Kennedy Class I RPDs, which replaced at least the second premolars and first molars were evaluated for possible inclusion in the present study. The existing RPDs were either attached to natural teeth, implants, or a combination of both. Further inclusion criteria included general health, a minimum age of 18 years, stable antagonizing dentition of natural or artificial teeth, a vertical bone quantity of at least 6 mm above the mandibular canal, and sufficient horizontal crest dimension for an implant diameter of 4 mm. The exclusion criteria included mental (e.g., alcohol or drug abuse), systemic (e.g., untreated diabetes mellitus), or local conditions (e.g., insufficient bone quantity). A detailed overview of eligibility criteria is given in the pilot of the present study [25].

After their inclusion to the study, patients were scheduled for the surgical procedure, comprising of the placement of two 6 mm short implants with a diameter of either 4.0 or 4.5 mm (SIC ace; SIC invent AG, Basel, Switzerland), placed bilaterally in mandibular molar sites. Surgery was defined as the baseline (BL) for the planning of subsequent follow-up visits. The desired implant site was the first molar position but when local defects were present, the implants were placed in second molar positions. The existing dentures served as a template for the determination of the implant position. The osteotomy and the implant placement were done according to manufacturer’s surgical protocol. All implants were placed at bone level, relative to the buccal bone crest, whenever possible, and cover screws were mounted, allowing for a submerged healing period of 3 months. If the vertical bone height was not sufficient to place the implant 6 mm deep and to maintain the safety distance to the alveolar nerve at the same time, the implant was placed in a position so that at least all threads were covered by bone (minimum insertion depth 4.5 mm). All implants were titanium grade 4 bone-level type implants with an integrated platform switch of 0.35 mm. The implant neck was non-threaded, and the first thread started 1.5 mm below the implant platform. Figure 1 illustrates the implant planning and the surgical procedures.

After a healing period of 3 months, the implants were uncovered and healing abutments were mounted. The denture was adapted in the implant sites, using a milling-cutter. One month later, an open-tray implant impression was taken, using the modified denture as an impression tray. A master cast was fabricated that was used for all subsequent denture modifications during the study. After an indirect relining of the dentures by a dental technician, the dome abutments were screwed on all implants, using a torque wrench with an insertion torque of 20 Ncm as recommended by the manufacturer, and the modified dentures were delivered (Figure 2a,b). The dome abutments were available in heights of 2 or 4 mm. The dome abutments were made of titanium grade 4, and had a semicircular shape without any retentive parts, similar to healing abutments for transgingival healing.

After wearing the dome abutments for two months, the abutments were removed and the dentures were modified again. Retentive ball abutments were mounted on the implants (Figure 3a–c). The ball had a diameter of 2.25 mm and the abutments were available in transgingival heights of either 2.0 or 4.0 mm (SIC invent AG, Basel, Switzerland). The retention of the corresponding matrices could be controlled with a screwdriver. All participants were scheduled for two additional monthly follow-up appointments after the abutment exchange. 

At the second follow-up appointment the participants were offered the opportunity to change the ball to the dome attachment, and if necessary, the abutments were exchanged, and the dentures relined. A final clinical follow-up visit was performed four months later (1 year after implant placement) (Figure 4).

OHRQoL was evaluated using the German version of the 49-item oral health impact profile (OHIP-G49) [26]. The questionnaires were filled by the participants after their inclusion into the study (before surgery), at the second stage surgery, and at 4, 8, 12, and 16 weeks after implant loading, and at the final visit 9 months after implant loading (i.e., 1 year after surgery). Each of the 49 items was assessed using a scale from 0 (happens never) to 4 (happens very often). The data were summed to obtain the global OHIP-score, from 0 to 196. The lower the score, the higher the OHRQoL. The seven subscales of OHIP, namely functional limitation, physical pain, psychosocial discomfort, physical disability, psychological disability, social disability, and handicap, were also calculated and compared. At the 16-week post-loading follow-up, an additional questionnaire with numerical rating scales (NRS) was given to the participants, asking for their preferred abutment. The NRS ranged from −10 (maximum preference for the dome abutment) to +10 (maximum preference for the ball abutment). A score of 0 represented no preference. The second question asked if the final abutment choice would have been different if there was a price difference of 500 Swiss Francs (CHF) between the two options (−10 absolutely yes, +10 absolutely not, 0 = maybe). At the final follow-up visit, the participants filled another NRS-based questionnaire, asking for the perceived changes on their oral situation, quality of life, and chewing ability, ranging from −10 (maximum worsening) to +10 (maximum improvement). A score of 0 represented no change. The final question asked whether the patients would choose the treatment they had received again in retrospect (−10 absolutely not, +10 absolutely yes, 0 = maybe). 

Mean OHIP scores and standard deviations, as well as median, minimum, and maximum OHIP scores were calculated. A linear regression analysis including patients as a random effect and an adjustment for the different time points was used to calculate the differences in the OHIP scores of the patients with the different abutment types. Cohen’s effect size (ES) was calculated with a 95%-confidence interval to analyze the difference of the two abutments relative to the baseline, as well as between the abutments in terms of OHIP scores. For the NRS based questionnaires, a one-sample *t*-test was performed to test whether the mean is equal to 0. All analyses were done with an α < 0.05.

## 3. Result

### 3.1. Study Sample

The initial study sample consisted of 19 finally included patients. As the data of 6 participants were already reported in the pilot study [25], the present study only considered data of 13 participants to avoid double-reporting. Among those participants, the youngest was 53 years old (age at implant placement) and the oldest 80 (median 68). There were 9 men (70%) and 4 women (30%). At baseline, oral hygiene ability, as measured by a 4-point scale was good in all patients. Most of the participants had had their last visit to a dentist between 1 to 6 months before the beginning of the study (median 3 months). Two participants had not seen a dentist for more than 2 years and one for 10 years. The period in which the participants had been wearing an RPD was highly variable, ranging from 9 months to 49 years (median 60 months).

### 3.2. Clinical and Patient Reported Outcomes

The success rate of the implants was 100%, according to Buser’s criteria [27]: the implants were in situ without any detectable mobility, there were no persistent complaints such as pain or foreign body sensation, as well as no peri-implant infection with putrid secretion and no persistent peri-implant radiotranslucency. More details about the clinical outcomes are reported in a separate publication [28]. At baseline, i.e., before surgery while wearing Kennedy Class I RPDs, patients had high OHIP summary and domain scores (Table 1).

After the second stage surgery and the connection of the dome abutment for 2 months, the OHIP scores decreased significantly in the summary score and most subscales when compared to baseline. No difference was found in the social and physical disability scales, or in the handicap scale (Table 2). After exchanging the abutments and wearing the ball abutments for 2 months, the OHIP score compared to baseline were significantly smaller in the OHIP summary scales, and most of the subscales (Table 2). No difference was found in the social disability scale. Comparing the two abutments directly, no significant differences could be observed (all *p* > 0.05). Neither the effect sizes for the subscales (max. ES: 0.42) nor for the summary scale (ES: −0.02) reached the level of a medium effect (Table 3). However, all participants finally chose the ball abutment. After 1 year, the OHIP summary score and most of the subscale scores were signficantly smaller compared to baseline (Table 1). No statistically significant change could be shown for the social disability scale (*p* = 0.567). The OHIP summary score decreased by 17.6 (95% CI: −27.1; −8.1) points (ES: −0.99; 95% CI: −1.88; −0.11).

The NRS demonstrated that the participants clearly preferred the ball over the dome abutments (Table 4). The participants indicated that their final decision would not change in case of a price difference of 500 CHF between the abutments (Table 4). The participants perceived a significant improvement in their oral situation, chewing ability, and quality of life (all *p* < 0.001) (Table 5). Similarly, all patients stated that they would do the treatment again if they could go back, documented by a minimum score of 9 out of 10. 

## 4. Discussion

The current study evaluated the impact of 6 mm short implants with two different abutment types, converting bilateral free-end RPDs to IARPDs, on OHRQoL and patient satisfaction. Implant placement and subsequent application of the non-retentive dome abutments lead to an increase of OHRQoL. Therefore, H-01 was rejected. No difference between the non-retentive dome and the retentive ball abutments was found. Therefore, the second 0-hypothesis (H-02) was not rejected. Nevertheless, all participants (*n* = 13) finally chose the ball abutment. 

Over the last two decades, OHRQoL has become a substantive tool to assess the quality and value of different dental treatments [18,19]. The OHIP is one of the most widely used assessment devices in the world. Due to its reproducibility and ease of use, it allows the monitoring of a treatment in terms of OHRQoL throughout cross-sectional or longitudinal studies and also comparisons with other studies [20,26]. Nevertheless, the comparison with other studies converting free-end RPDs into IARPDs is challenging, as the versions or methods of administration of OHIP questionnaires are not always the same and there is a lack of reference values depending on the baseline conditions (e.g., type of edentulism, duration of edentulism, replacement of missing teeth, etc.) [23,24].

The baseline OHIP score of the study participants was high, attesting their inconveniences with their Kennedy class I RPDs, especially in terms of physical pain and functional limitation such as reduced masticatory performance. It should be emphasized that this score is quite a bit lower than those from other studies [16,21,29]. Nevertheless, the improvement demonstrated by the OHIP scores after the transformation of the RPDs was very similar compared to other studies [26]. The scores of the present study seem to be slightly more favourable than those and another of a Dutch study [29], which obtained a mean score of 16.1 after conversion from Kennedy class I to implant-supported Kennedy class III dentures. However, the present study lasted only a few months, while in the mentioned study, patients were followed over 16 years.

In recent years, various studies have investigated methodological considerations and interpretations of the evolution of OHIP scores [19,30]. In addition to the *p*-value, which serves as a mathematical reference for assessing whether a change is statistically significant or not, the concept of the minimal important difference (MID) is becoming increasingly meaningful [30]. This is the minimal score difference considered to be clinically relevant. For the OHIP-49 questionnaire, it is widely accepted that the MID for the summary score is 6 units [31]. Considering an MID of 6 points, the evolution of the OHIP scores demonstrated in the present study can be considered relevant, although not all subscale scores showed statistically significant changes. Consequently, both abutment types can be recommended to improve OHRQoL in Kennedy Class I RPD wearers. 

Both, dome and ball attachments lead to significant improvement of the participants’ OHRQoL. Although the difference in OHIP scores is minimal between the two treatment options, patients strongly preferred the ball abutment. No patients finally decided to keep the dome abutment. However, it should be considered that the abutment sequence in the present study was not randomized and that all patients received the dome attachment first and then the ball attachment. Thus, a possible carry-over effect cannot be excluded [32]. Therefore, the improvement induced by the dome attachment could have positively influenced the patients’ feelings about the ball attachment that followed. Therefore, the improvement induced by the dome attachment could have influenced the patients’ feelings about the ball attachment that followed in a positive way. Another reason for choosing the ball abutment could be that it did not require another modification of the dentures, as the patients were already wearing this abutment type. To eliminate this bias, a randomized crossover study could provide more reliable answers. However, the abutment sequence in the present study was selected based on the data of the pilot randomized crossover study. In the pilot study, only 2 out of 12 patients finally chose the dome abutment [25]. Other studies have also shown better clinical outcomes and improved patient satisfaction with retentive abutments [33,34]. However, it was also demonstrated that the use of vertical stops may work and provide high satisfaction, if the RPDs already had sufficient retention and adequate tooth abutments [33].

Generally, ES of >0.5 as a threshold for a medium effect of a specific treatment and ES > 0.8 are considered as a large effect [35]. The ES of the OHIP summary score after completing the treatment (ES: 0.99) indicates a large effect of placing two implants in the posterior mandible, converting a Kennedy Class I RPD to an IARPD. The most obvious differences in OHIP scores between the ball and the dome abutment were found in the fields of physical pain and handicap with an ES of 0.35 and 0.42, respectively. Even in those OHIP subscales, the ES did not reach the threshold for a medium influence of the abutment on the treatment outcome, indicating a minor influence of the abutment choice on the posterior implants. Furthermore, the subcategory handicap is difficult to compare in the context of prosthetic treatments because the value is often low [36]. 

It should be kept in mind that one of the main complaints of RPD wearers is the presence of food remnants under the denture [37]. Stabilizing the RPD, using a retentive abutment stabilizes the denture and may therefore reduce the amount of food remnants under the dentures. This factor, as well as the lower OHIP-49 total score for ball-abutment could therefore explain the patients’ preference for the attachment offering additional retention, although no significant difference between the abutments was detected. This would explain the participants’ willingness to pay CHF 500 more for the ball abutment. Indeed, it was calculated that the transformation to IARPD was suitable if the patient was willing to pay at least 80 EUR more per OHIP point won [38]. In general, the participants were satisfied with the transformation of their RPDs to IARPDs. The NRS questionnaires showed significant improvement of their oral situation, quality of life and chewing ability. Other studies have also shown improvement not only in overall satisfaction, but also in stability, chewing and appearance [39], or in retention, comfort, masticatory capacity, and speaking ability [40].

Short implant placement to support bilateral mandibular Kennedy class I RPDs showed both statistically significant and clinically relevant improvement. For instance, all participants of the current study would have the treatment again. However, the limitations of the present study, including the small sample size, missing sample size analysis, short follow-up period, and missing control group must be considered. Additional research is needed to establish criteria for choosing the ideal abutment type on short posterior implants.

## 5. Conclusions

Considering the above-mentioned limitations, short implants in posterior mandibular sites, transforming RPDSs to IARPDs can be recommended to improve OHRQoL. Patients seem to prefer retentive over non-retentive abutments. 

## Figures and Tables

**Figure 1 ijerph-19-08998-f001:**
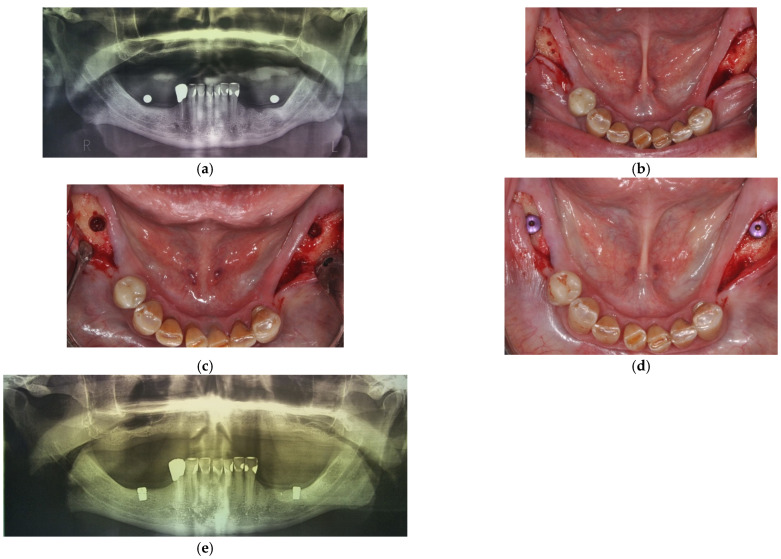
Surgical planning and implant placement. (**a**) Initial orthopantomogram (OPT) for implant planning, including metal balls for linear measurements; (**b**) preparation of the mucoperiosteal flap; (**c**) intraoral situation after the osteotomy; (**d**) implants with cover screws inserted; (**e**) post-surgical OPT.

**Figure 2 ijerph-19-08998-f002:**
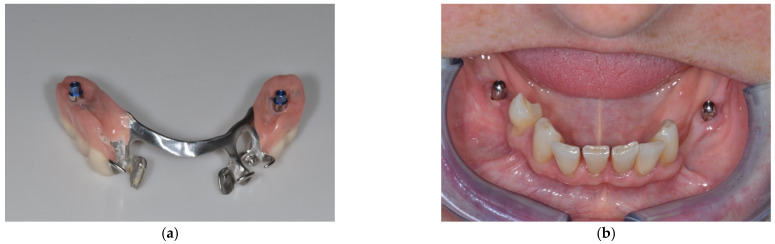
(**a**) Extraoral view of the converted prosthesis with dome abutments in place; (**b**) intraoral view with dome abutments mounted on the implants.

**Figure 3 ijerph-19-08998-f003:**
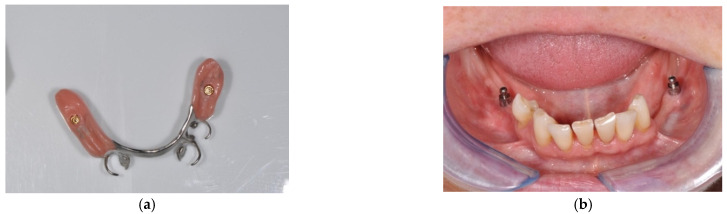
(**a**) Extraoral view of the converted prosthesis including the matrices for the ball abutments; (**b**) intraoral view with ball abutments mounted on the implants; (**c**) intraoral view of the converted implant-assisted removable partial denture.

**Figure 4 ijerph-19-08998-f004:**
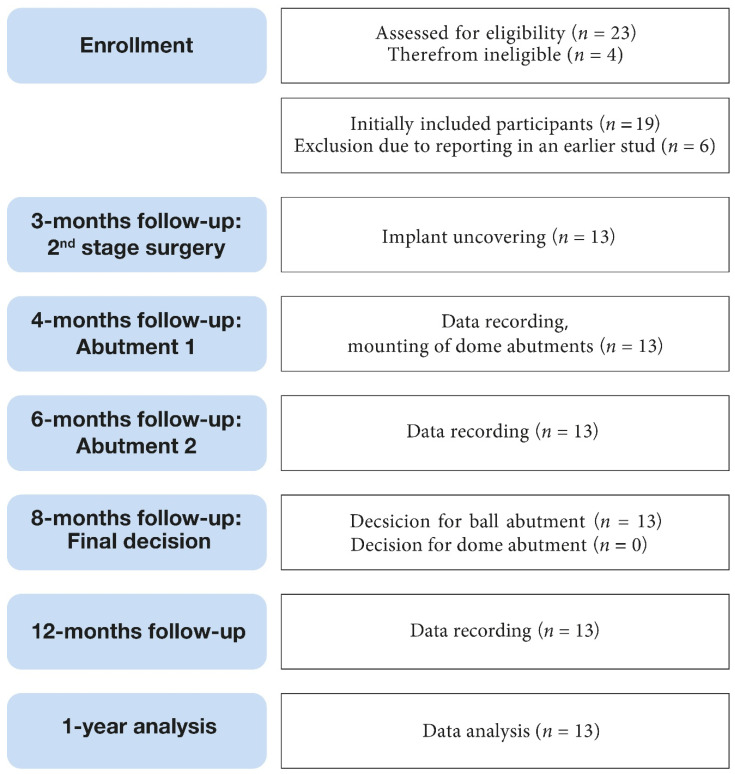
Flow-chart summarizing the study procedures.

**Table 1 ijerph-19-08998-t001:** Oral Health Impact Profile (OHIP) at baseline and after 12 months.

		Mean	Sd	Median	Min–Max	Diff [95% CI] Effect Size [95% CI]	*p*-Value
*Functional limitation*	BL	8.2	4.0	8.0	3.0–15.0		
	12 months	4.0	3.4	3.0	0.0–10.0		
	Diff	−4.2	4.2	−3.0	−12.0–2.0	−4.2 [−6.6; −1.9]	<0.001
						−0.89 [−1.70; −0.08]	
*Physical pain*	BL	7.8	6.8	7.0	0.0–26.0		
	12 months	1.7	1.9	2.0	0.0–6.0		
	Diff	−6.1	7.9	−6.0	−26.0–6.0	−6.1 [−10.8; −1.3]	0.012
						−0.93 [−1.80; −0.05]	
*Psychological discomfort*	BL	2.8	3.4	1.0	0.0–11.0		
	12 months	1.0	1.9	0.0	0.0–6.0		
	Diff	−1.8	2.8	−1.0	−6.0–3.0	−1.8 [−3.4; −0.3]	0.020
						−0.62 [−1.40; 0.17]	
*Physical disability*	BL	5.2	4.6	5.0	0.0–13.0		
	12 months	2.0	3.0	0.0	0.0–8.0		
	Diff	−3.2	3.1	−2.0	−9.0–0.0	−3.2 [−5.0; −1.5]	<0.001
						−0.73 [−1.52; 0.07]	
*Psychological disability*	BL	2.5	3.1	0.0	0.0–8.0		
	12 months	0.9	1.7	0.0	0.0–5.0		
	Diff	−1.6	2.5	0.0	−6.0–1.0	−1.6 [−3.0; −0.2]	0.021
						−0.60 [−1.38; 0.19]	
*Social disability*	BL	0.8	1.0	1.0	0.0–3.0		
	12 months	0.6	1.3	0.0	0.0–4.0		
	Diff	−0.2	1.4	0.0	−3.0–3.0	−0.2 [−1.0; 0.6]	0.567
						−0.20 [−0.97; 0.57]	
*Handicap*	BL	2.6	2.3	2.0	0.0–8.0		
	12 months	1.6	2.6	0.0	0.0–7.0		
	Diff	−1.0	1.6	−1.0	−3.0–1.0	−1.0 [−1.9; −0.1]	0.025
						−0.40 [−1.17; 0.38]	
* OHIP-G49 *	BL	27.5	14.1	30.0	4.0–50.0		
* TOTAL SCORE *	12 months	9.9	11.0	5.0	0.0–35.0		
	Diff	−17.6	15.7	−19.0	−50.0–5.0	−17.6 [−27.1; −8.1]	<0.001

OHIP scores at baseline and after 12 months; differences and *p*-values result from the linear regression adjusted for the follow-up period.

**Table 2 ijerph-19-08998-t002:** Difference (Diff) of dome and ball attachments vs. baseline (BL).

* Subscale *	Diff Dome→BL	Diff Ball→BL
*Functional limitation*	−2.6 (Sd 4.6), *p* = 0.018	−3.5 (Sd 4.3), *p* < 0.001
*Physical pain*	−5.4 (Sd 7.3), *p* < 0.001	−5.4 (Sd 8.3), *p* < 0.001
*Psychological discomfort*	−1.7 (Sd 2.8), *p* = 0.006	−2.0 (Sd 2.9), *p* < 0.001
*Physical disability*	−1.5 (Sd 4.1), *p* = 0.125	−3.1 (Sd 3.2), *p* < 0.001
*Psychological disability*	−1.2 (Sd 2.4), *p* < 0.001	−1.7 (Sd 2.6), *p* < 0.001
*Social disability*	−0.1 (Sd 1.5), *p* = 0.674	−0.1 (Sd 1.4), *p* = 0.820
*Handicap*	−1.1 (Sd 2.6), *p* = 0.068	−1.0 (Sd 1.6), *p* = 0.039
* OHIP-G49 TOTAL *	−13.2 (Sd 17.7), *p* = 0.002	−15.8 (Sd 15.8), *p* < 0.001

Difference dome vs. baseline (after 6 months) and ball vs. baseline (after 8 months).

**Table 3 ijerph-19-08998-t003:** Difference and effect size (ES) between the two attachments.

* Subscale *	Diff Dome VS. Ball	Cohen’s ES	*p*-Value
*Functional limitation*	−0.7 [−3.8; 2.5]	−0.15 [−0.72; 0.41]	0.670
*Physical pain*	2.6 [−0.8; 6.1]	0.35 [−0.23; 0.92]	0.137
*Psychological discomfort*	0.2 [−0.7; 1.0]	0.06 [−0.51; 0.64]	0.675
*Physical disability*	−0.5 [−4.4; 3.4]	−0.14 [−0.71; 0.44]	0.798
*Psychological disability*	−0.3 [−2.0; 1.4]	−0.12 [−0.69; 0.45]	0.720
*Social disability*	−0.3 [−2.0; 1.4]	−0.12 [−0.69; 0.45]	0.720
*Handicap*	0.9 [−1.1; 2.8]	0.42 [−0.16; 0.99]	0.375
* OHIP-G49 TOTAL *	−0.3 [−9.3; 8.8]	−0.02 [−0.61; 0.57]	0.953

Difference in OHIP between dome at and ball attachment after wearing both attachments for two months; negative values indicate a higher effect of the ball.

**Table 4 ijerph-19-08998-t004:** Participant ratings at study termination: Numerical rating scales.

	Mw	Sd	Median	Min–Max	*p*-Value
Which abutment would you choose? (−10 preference for the dome/+10 preference for the ball)	8.4	3.2	10.0	1.0–10.0	<0.001
Would your answer be different if there was a difference of 500 CHF? (yes 0/no +10)	9.7	0.7	10.0	8.0–10.0	<0.001

Participants indicated a clear preference for the ball abutment. Abbreviations: Mw = mean, Sd = standard deviation

**Table 5 ijerph-19-08998-t005:** Perceived changes at study termination: Numerical rating scales.

	Mean	Sd	Median	Min–Max	*p*-Value
Oral situation	7.9	3.2	9.0	0.0–10.0	<0.001
Quality of life	8.5	2.6	9.5	0.0–10.0	<0.001
Chewing ability	8.5	2.3	9.5	2.0–10.0	<0.001
Treatment again	9.9	0.2	10.0	9.0–10.0	<0.001

Participants showed high satisfaction scores at study termination.

## Data Availability

The data presented in this study are available on request from the corresponding author. The data are not publicly available due to ethic matters.

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
