# Peer review of "Converting Bilateral Free-End Removable Partial Dentures to Implant-Assisted Removable Partial Dentures Using 6 mm Short Implants: Patient-Reported Outcomes of a Prospective Clinical Study"

_ijerph, 2022, doi:10.3390/ijerph19158998_

Round 1

Reviewer 1 Report

I should congratulate the authors for writing a clean and an efficient paper. If I may advice to authors; their paper will be more efficient, they will insert 2 pictures for showing abutments they used.

Author Response

I should congratulate the authors for writing a clean and an efficient paper. If I may advice to authors; their paper will be more efficient, they will insert 2 pictures for showing abutments they used.

Response: Thank you for your kind words. We added clinical images to show the different abutment types.

English language and style are fine/minor spell check required 

Response: Style and spell checking have been done.

Reviewer 2 Report

 The manuscript lacks power analysis and sample size calculation. There is also no CONSORT flow chart in the manuscript showing participant  recruitment and follow up.

Author Response

The manuscript lacks power analysis and sample size calculation. There is also no CONSORT flow chart in the manuscript showing participant recruitment and follow up.

Response: Thank you for mentioning this important issue. The small number of participants was already mentioned as a limitation. However, following your advice, we have added the missing sample size calculation as a further weak point. Furthermore, a study flow-chart has been added.

Reviewer 3 Report

Very interesting paper and very nice presentation.

However, I think the reader would like to see a series of pictures of the cases that were converted to an implant-assisted removable partial dentures.

Specifically, I would like to see a case from the authors that they have converted a free-end partial-denture to implant assisted.

Moreover, I would like to see the initial situation, the planning of the implant case and the steps of transformation of the free-end denture.

Author Response

Reviewer 3

Very interesting paper and very nice presentation.

Response: Thank you so much for kindly evaluating our study

However, I think the reader would like to see a series of pictures of the cases that were converted to an implant-assisted removable partial dentures. Specifically, I would like to see a case from the authors that they have converted a free-end partial-denture to implant assisted.

Response: We added clinical images from a case that has been converted from a bilateral free-end to an implant assisted removable partial denture, showing both abutment types. Please note, that all prior publications already include images that you requested. Please see the references:

Enkling N, Nauli J, Kraus D, Wittneben JG, Schimmel M, Abou-Ayash S. Short strategic implants for mandibular removable partial dentures: One-year results from a pilot randomized crossover abutment type study. Clin Oral Implants Res. 2021 Oct;32(10):1176-1189. doi: 10.1111/clr.13815. Epub 2021 Aug 25. PMID: 34352145; PMCID: PMC9292160.

Enkling N, Thanendrarajah T, Mathey A, Janner S, Schimmel M, Abou-Ayash S. Soft Loading Protocol of Short Strategic Implants in Posterior Mandibles Supporting Removable Bilateral Free-End Prostheses: 1-Year Results of a Prospective Cohort Study. Int J Prosthodont. 2022 Feb 22. doi: 10.11607/ijp.7827.

Moreover, I would like to see the initial situation, the planning of the implant case and the steps of transformation of the free-end denture.

Response: An OPT showing the initial situation of one participant as well as the post-surgical OPT have been added to the manuscript. Furthermore, we added clinical images of the surgery and the final situation